# p53-Dependent Apoptotic Effect of Puromycin via Binding of Ribosomal Protein L5 and L11 to MDM2 and Its Combination Effect with RITA or Doxorubicin

**DOI:** 10.3390/cancers11040582

**Published:** 2019-04-24

**Authors:** Ji Hoon Jung, Hyemin Lee, Ju-Ha Kim, Deok Yong Sim, Hyojin Ahn, Bonglee Kim, Suhwan Chang, Sung-Hoon Kim

**Affiliations:** 1College of Kyung Hee Medicine, Kyung Hee University, Seoul 02447, Korea; johnsperfume@khu.ac.kr (J.H.J.); glansy555@gmail.com (H.L.); 964juha@daum.net (J.-H.K.); simdy0821@naver.com (D.Y.S.); hyojin1982@gmail.com (H.A.); bongleekim@khu.ac.kr (B.K.); 2Department of Biomedical Sciences, College of Medicine, University of Ulsan, Asan Medical Center, Seoul 05505, Korea; suhwan.chang@amc.seoul.kr

**Keywords:** Puromycin, RPL5, RPL11, p53, doxorubicin

## Abstract

Among ribosomal proteins essential for protein synthesis, the functions of ribosomal protein L5 (RPL5) and RPL11 still remain unclear to date. Here, the roles of RPL5 and RPL11 were investigated in association with p53/p21 signaling in the antitumor effect of puromycin mainly in HCT116 and H1299 cancer cells. Cell proliferation assays using 3-[4,5-dimethylthiazole-2-yl]-2,5-diphenyltetrazolium bromide (MTT) assays and colony formation assays, cell cycle analysis, Reverse transcription polymerase chain reaction (RT-PCR) and Western blotting were performed in cancer cells. Puromycin exerted cytotoxic and anti-proliferative effects in p53 wild-type HCT116 more than in p53 null H1299 cells. Consistently, puromycin increased sub-G1, cleaved Poly (ADP-ribose) polymerase (PARP), activated p53, p21, and Mouse double minute 2 homolog (MDM2), and attenuated expression of c-Myc in HCT116 cells. Notably, puromycin upregulated the expression of RPL5 and RPL11 to directly bind to MDM2 in HCT116 cells. Conversely, deletion of RPL5 and RPL11 blocked the activation of p53, p21, and MDM2 in HCT116 cells. Also, puromycin enhanced the antitumor effect with reactivating p53 and inducing tumor apoptosis (RITA) or doxorubicin in HCT116 cells. These findings suggest that puromycin induces p53-dependent apoptosis via upregulation of RPL5 or RPL11 for binding with MDM2, and so can be used more effectively in p53 wild-type cancers by combination with RITA or doxorubicin.

## 1. Introduction

Puromycin, an old antibiotic derived from *Streptomyces alboniger* as a structural analog of tyrosyl tRN 1 [1], is known to induce apoptosis in breast cancer cells by insulin-like growth factor 1 (IGF-I) and exert antitumor activity in MDA-MB-231 cells via the suppression of 45S pre-ribosomal RNA and upstream binding factor (UBF) [2,3], since it terminates the ribosomal protein synthesis process by causing the premature release of a polypeptide from the ribosome in malignant cells compared to normal cells [4]. Recently, many puromycin derivatives have been developed for clinical applications [5]. The cellular processes such as development, differentiation, cell proliferation, and apoptosis are controlled indirectly or directly by oncogenes and tumor suppressors including c-Myc, PTEN, and p53 [6,7]. The p53 tumor suppressor protein is a major mediator of cell-cycle arrest and/or apoptosis in the response of mammalian cells to cellular stress, including nucleolar stress or ribosomal stress [8]. In addition, p53 signaling is inactivated by two important regulators of p53—mouse double minute 2 (MDM2) [9] and MDMX (also known as HDMX and MDM4), by their ubiquitin dependent degradation of p53 [10,11]. Emerging evidence reveals that the disruption of ribosome biogenesis and/or the nucleolar structure activates p53-dependent or independent signaling pathways leading to cell cycle arrest, apoptosis, differentiation, and senescence [12,13]. Some ribosomal proteins in particular, including RPL5, RPL11, and RPS14 have been reported to regulate p53 expression in several cancer cells [14,15,16,17]. Also, inhibition of ribosomal RNA processing and ribosomal stress was found to activate p53 signaling [18,19,20]. Nevertheless, the underlying antitumor mechanism of puromycin has not been explored in association with ribosomal proteins and p53/MDM2 signaling so far. Thus, in the present study, the roles of RPL5 and RPL11 were elucidated in association with p53/MDM2 signaling in the puromycin-induced antitumor effect in p53 sensitive and deficient cancer cells. 

## 2. Results

### 2.1. Puromycin Exerts Cytotoxic and Antiproliferative Effects in Cancer Cells

To evaluate the cytotoxic and antiproliferative effects of puromycin, an 3-[4,5-dimethylthiazole-2-yl]-2,5-diphenyltetrazolium bromide (MTT) assay and colony formation assay were adopted in several cancer cells. Here, puromycin significantly suppressed the viability of p53 wild-type HCT116 cells in a concentration and time-dependent fashion compared to SW620, HCT15, and H1299 cells using an MTT assay (Figure 1A,B). Similarly, puromycin inhibited the proliferation of HCT116 cells (p53 wild type), not H1299 cells (p53 null type) by a colony formation assay (Figure 1C). 

### 2.2. Puromycin Regulates Apoptosis-Related Proteins along with Increase of Sub-G1 Population

To confirm whether cytotoxic and antiproliferative effects of puromycin are due to cell cycle arrest and apoptosis, cell cycle analysis and Western blotting were performed in HCT116 and/or H1299 cells. Here, puromycin increased the sub-G1 population in HCT116 cells (Figure 2A). However, normal colon cells (CCD-18co) were not affected by puromycin (Figure 2B). In addition, Western blotting showed that the expression of cyclin D1 and CDK4 for G1-S transition was reduced in a concentration and time-dependent manner (Figure 2C,D). In addition, puromycin significantly cleaved Poly (ADP-ribose) polymerase (PARP) and attenuated the expression of Bcl-xL, Bcl-2, and phopho-AKT in a concentration-dependent manner in p53 wild-type HCT116 cells better than in p53 null H1299 cells (Figure 2E,F).

### 2.3. Puromycin Enhances p53 Stability along with Inhibition of c-Myc in HCT116 Cells

It is well documented that p53 plays an important role in cancer cells as a tumor suppressor [21] and is closely associated with c-Myc downregulation [22]. Herein, puromycin significantly upregulated the expression of p53 and p21 and suppressed c-Myc at the protein level, and also activated mRNA expression of p21 in HCT116 cells (Figure 3A,B). Of note, puromycin maintained p53 stability despite protein synthesis inhibition by protein synthesis inhibitor cycloheximide (CHX) (Figure 3C). Additionally, the depletion of p53 using small interfering RNA (siRNA) transfection reduced PARP cleavages and p21 activation by puromycin in HCT116 cells (Figure 3D).

### 2.4. Ribosomal Protein L5 and L11 Mediate p53 Activation via Direct Binding to MDM2 in HCT116 Cells—An Example of an Equation

It is well known that ribosomal biogenesis and translation are regulated by the 5S rRNA and tRNA genes [23]. To determine whether puromycin affects ribosomal biogenesis and translation, Reverse transcription polymerase chain reaction (RT-PCR) analysis was performed in HCT116 cells. As shown in Figure 4A, the mRNA levels of tRNATyr, tRNALeu, and nucleolin were significantly decreased compared to the untreated control. Several ribosomal proteins including L5, L11, and L23 directly interact with MDM2 to block p53 ubiquitination mediated by MDM2 [24]. Notably, depletion of RPL5 or RPL11 blocked activation of p53/p21 and MDM2 compared to the untreated control (Figure 4B,C). Consistently, the GeneCards program (STRING V10.5) showed the interaction scores of RPL5 or RPL11 with MDM2 to be 0.962 and 0.980, respectively, compared to p53 (Figure 4D). Furthermore, direct binding between MDM2 and RPL5 or RPL11 was confirmed in puromycin-treated HCT116 cells by immunoprecipitation (Figure 4E,F).

### 2.5. Combination Effect of Puromycin and Reactivating p53 and Inducing Tumor Apoptosis (RITA) in HCT116 Cells as Antitumor Agents

To assess the combination of puromycin and RITA, an MTT assay was conducted in HCT116 cells. Here, puromycin enhanced the cytotoxicity of RITA in a concentration-dependent manner in HCT116 cells (Figure 5A). Furthermore, puromycin induced PARP cleavage and inhibited Bcl-2 and cyclin D1 with RITA in p53 wild-type HCT116 cells more than in p53 null H1299 cells (Figure 5B). Consistently, combination with puromycin and doxorubicin promoted more effective apoptosis by attenuating c-Myc and Bcl-2 and activating p53 in p53 wild-type HCT116 cells more than in p53 null H1299 cells (Figure 5C,D).

## 3. Discussion

Although puromycin has not been used for cancer therapy as a peptidase inhibitor [25] or a tyrosine kinase inhibitor [26] to date, puromycin derivatives have been used for cancer treatment [25,27]. Thus, for the possibility of the application of puromycin in cancer therapy, in the current study, the underlying apoptotic mechanism of puromycin was investigated in association with RPL5 or RPL11-mediated p53 activation signaling. Herein, puromycin treatment induced significant cytotoxicity in p53 wild-type HCT116 cells more than in p53 mutant or null SW620, HCT15 colon cancer cells, and H1299 non-small lung cancer cells by MTT assay, implying potent involvement of the p53 signaling pathway in the antitumor effect of puromycin. Likewise, puromycin suppressed the proliferation of HCT116 cells (p53 wild-type), not p53 mutant SW620 or HCT15 and H1299 cells (p53 null-type) by colony formation assay. Notably, H1299 cells (p53 null-type) were more susceptible to puromycin than p53 mutant HCT15 colon cancer cells, which may be attributed to cell-specific different gene characterization. 

To confirm whether the cytotoxicity of puromycin is due to apoptosis, cell cycle analysis was conducted. Here, puromycin increased the sub-G1 population in HCT116 cells. Consistently, Western blotting showed the reduced expression of cyclin D1 and CDK4 for G1-S transition in puromycin-treated HCT116 cells, indicating the apoptotic potential of puromycin in HCT116 cells. Furthermore, puromycin significantly cleaved PARP as an apoptotic feature and attenuated the expression of antiapoptotic proteins such as Bcl-xL, Bcl-2, and phopho-AKT in a concentration-dependent manner in p53 wild-type HCT116 cells better than in p53 null H1299 cells, possibly indicating the p53-dependent apoptosis of puromycin. 

Accumulating evidence showed that the tumor suppressor p53 as a transcription factor is repressed by MDM2 or MDM4, and this promotes p53 degradation via ubiquitination only under non-activated conditions [28,29,30]. Additionally, recent evidence demonstrated that endogenous c-Myc efficiently induced p53-dependent apoptosis following DNA damage [22,31]. Western blotting showed that puromycin significantly activated p53/p21 at protein and mRNA levels and also suppressed c-Myc in a time-dependent fashion in HCT116 cells, demonstrating that puromycin is an activator of p53/p21. Consistently, puromycin maintained p53 stability in HCT116 cells, whereas DNA synthesis inhibitor CHX attenuated the expression of p53. 

Emerging evidence revealed that the dysfunction of c-Myc [22] and its regulated ribosomal biogenesis by ribosomal RNA synthesis and processing are critically involved in p53 activation [13,32]. Here, puromycin suppressed the mRNA levels of tRNATyr, tRNALeu, and nucleolin as c-Myc regulated RNA Pol-transcribed genes compared to the untreated control, implying potent regulation of RNA Pol I and II target genes by puromycin. In addition, it is well known that RPL5 [33], RPL11 [32,34], and RPL23 [35] target MDM2 and p53 in response to ribosomal stress [12,24], since these proteins directly interact with MDM2 to block p53 ubiquitination mediated by MDM2 [29,36,37]. Immunoprecipitation demonstrated that RPL5 or RPL11 directly binds to MDM2 in HCT116 cells, and this was supported by a binding score between RPL5 or RPL11 with MDM2 from the GeneCards program. Conversely, depletion of RPL5 or RPL11 blocked activation of p53/p21 and MDM2 by puromycin compared to in the untreated control, indicating the pivotal role of RPL5 or RPL11 in puromycin-induced apoptosis in p53 wild-type HCT116 cells. 

Additionally, puromycin enhanced the antitumor effect in HCT116 cells with an MDM2 inhibitor or p53 activator RITA by MTT assay. Likewise, puromycin promoted p53 activation, PARP cleavage, and inhibition of Bcl-2 and cyclin D1 induced by RITA in p53 wild-type HCT116 cells more than in p53 null H1299 cells, demonstrating the combination effect of puromycin and RITA. Furthermore, puromycin enhanced the inhibition of c-Myc and Bcl-2 and activation of p53 by doxorubicin in HCT116 cells, suggesting a potent combination therapy of puromycin and doxorubicin especially in p53 wild-type cancers, though further study is required in animal studies and human clinical trials in the future. 

## 4. Materials and Methods

### 4.1. Cell Culture

HCT116 (ATCC^®^ CCL-247™, p53 wild-type human colon cancer), SW620 (ATCC^®^ CCL-227™, p53 mutant human colon cancer), HCT15 (ATCC^®^ CCL-225™, p53 mutant human colon cancer), and H1299 (ATCC^®^ CRL-5803™, p53 null-type human non-small cell lung cancer) were purchased from American Type Culture Collection (ATCC, Manassas, VA, USA), and were maintained in a RPMI1640 medium supplemented with 10% fetal bovine serum (FBS), 2 μM L-glutamine and penicillin/streptomycin at 37 °C in a humidified incubator containing 5% CO_2_. 

### 4.2. Cytotoxicity Assay

The cytotoxicity of puromycin was tested with a 3-(4,5-dimethylthiazol-2-yl)-2,5-diphenyltetrazolium bromide (MTT) (Sigma Aldrich Co., St. Louis, MO, USA) assay. HCT116, SW620, HCT15, and H1299 cells were seeded onto 96-well plates (1 × 10^4^ cells/well) and exposed to various concentrations of puromycin (0, 0.063, 0.125, 0.25, 0.5, and 1 μg/mL) for 24 h or 48 h. The puromycin was purchased from Sigma (P8833). The cells were incubated with MTT solution (1 mg/mL) until formazan was constituted. The optical density was measured at 570 nm using a microplate reader (TECAN, Grödig, Austria). The cell viability was calculated by the following equation: cell viability (%) = [OD (puromycin) − OD (blank)]/[OD (control) − OD (blank)] × 100.

### 4.3. Cell Cycle Analysis

HCT116 cells were exposed to puromycin (0.25 and 0.5 μg/mL) for 24 h. The cells were washed and fixed in 70% ethanol, incubated with 0.1% RNAse A in phosphate-buffered saline at 37 °C for 30 min and resuspended in phosphate-buffered saline containing 25 μg/mL propidium iodide in the dark. The DNA contents of the stained cells were analyzed using CellQuest software (BD Biosciences, San Jose, CA, USA) with FACSCalibur flow cytometry (Becton Dickinson, Franklin Lakes, NJ, USA).

### 4.4. RNA Interference and Inhibitor Study

HCT116 cells were transfected with siRNA plasmids using INTERFERin^®^ (Polyplus-transfection SA, Illkirch, France) transfection reagent according to the manufacturer’s protocol. Additionally, HCT116 cells treated with puromycin for 24 h were exposed to 50 μg/mL cycloheximide as a protein synthesis inhibitor for 30, 60, and 120 min before harvesting to assess p53 stability by Western blotting. 

### 4.5. Colony Formation Assay

The cells were seeded onto 60 mm plates at a density of 1000 cells/well, and then were treated by puromycin for 24 h. A week later, the cells were rinsed with PBS twice, fixed with 100% methanol for 30 min at room temperature, and stained with 0.5% crystal violet for 1 h at room temperature. Stained cells were solubilized with 1 mL of 1% SDS buffer at room temperature (RT) for 30 min and the lysates were transferred onto 96-well plates. Then the optical density was measured using a microplate reader (Sunrise, TECAN, Switzerland) at a 590 nm wavelength. 

### 4.6. Western Blot and Co-Immunoprecipitation Analysis

The cells were lysed in radioimmunoprecipitation assay (RIPA) buffer (0.1% SDS, 1 mM EDTA, 50 mM Tris-HCl, pH 7.4, 150 mM NaCl, 1% Triton X-100, 1 mM Na3VO4, 1 mM NaF, protease inhibitor cocktail). Cell lysates were separated to SDS-PAGE (8–15%) by electrophoresis and transferred onto a Hybond enhanced chemiluminescence (ECL) transfer membrane (Amersham Pharmacia, Piscataway, NJ, USA). After blocking in 3–5% skim milk, the membranes were probed with multiple antibodies against c-Myc (1:2000), RPL5 (1:1000), RPL11 (1:1000) (Abcam, Cambridge, United Kingdom), phospho-AKT (Ser473) (1:1000), AKT (1:1000), PARP(1:1000), Bcl-2 (1:1000), Bcl-x_L_ (1:1000), cyclin D1 (1:1000), CDK4 (1:1000), p53 (DO-1) (1:2000), MDM2 (Santa Cruz Biotechnologies, Santa Cruz, CA, USA) (1:500), p21 (Ab frontier, Seoul, Korea) (1:500), and β-actin (Sigma Aldrich Co., St. Louis, MO, USA) (1:3000) and exposed to horseradish peroxidase (HRP)-conjugated secondary anti-mouse or rabbit antibodies (1:5000) for 2 h. Protein expression was measured using the ECL system (Amersham Pharmacia, Piscataway, NJ, USA). Immunoprecipitation was performed using RPL5 or RPL11 and MDM2 antibodies. Then, protein G or A beads (Santa Cruz Biotechnologies, Santa Cruz, CA, USA) were washed three times with lysis buffer. The final precipitated proteins were subjected to immunoblotting.

### 4.7. Real-Time Quantitative PCR Analysis (RT-qPCR)

Total RNAs were isolated using the RNeasy mini kit (Qiagen, Valencia, CA, USA) according to the manufacturer’s instructions and reverse transcribed using M-MLV reverse transcriptase (Promega, Madison, WI, USA). The cDNAs were amplified by PCR using the synthesized specific primers (Bioneer, Daejeon, Korea) (Table 1). 

RT-qPCR was operated with the lightcycler TM instrument (Roche Applied Sciences, Indianapolis, IN, USA) according to the manufacturer’s protocol. PCR started at 95 °C for 10 min, followed by 40 cycles of 95 °C for 10 s, 57 °C for 15 s, and 72 °C for 20 s. The mRNA level of glyceraldehyde-3-phosphate dehydrogenase (GAPDH) was used to normalize the expression of genes of interest. Each sample was tested in triplicate, and relative gene expression data were analyzed by means of the 2^−ΔCT^ method.

### 4.8. Statistical Analysis

Data were presented as means ± standard deviation (SD). The statistically significant differences between the control and puromycin-treated groups were calculated using Student’s *t*-test. All experiments were carried out in triplicate.

## 5. Conclusions

Our findings suggest that puromycin exerted significant cytotoxic and anti-proliferative effects and regulated apoptotic proteins such as PARP, BcL-2, p-AKT, c-Myc, p53/p21 in p53 wild-type HCT116 cells better than in p53 null H1299 cells. Notably, RPL5 or RPL11 critically regulated p53-dependent apoptosis by puromycin in HCT116 cells. In addition, puromycin enhanced the antitumor effect with RITA or doxorubicin. Overall, these findings suggest that puromycin can be used effectively in p53 wild-type cancers with a potent combination of RITA or doxorubicin. 

## Figures and Tables

**Figure 1 cancers-11-00582-f001:**
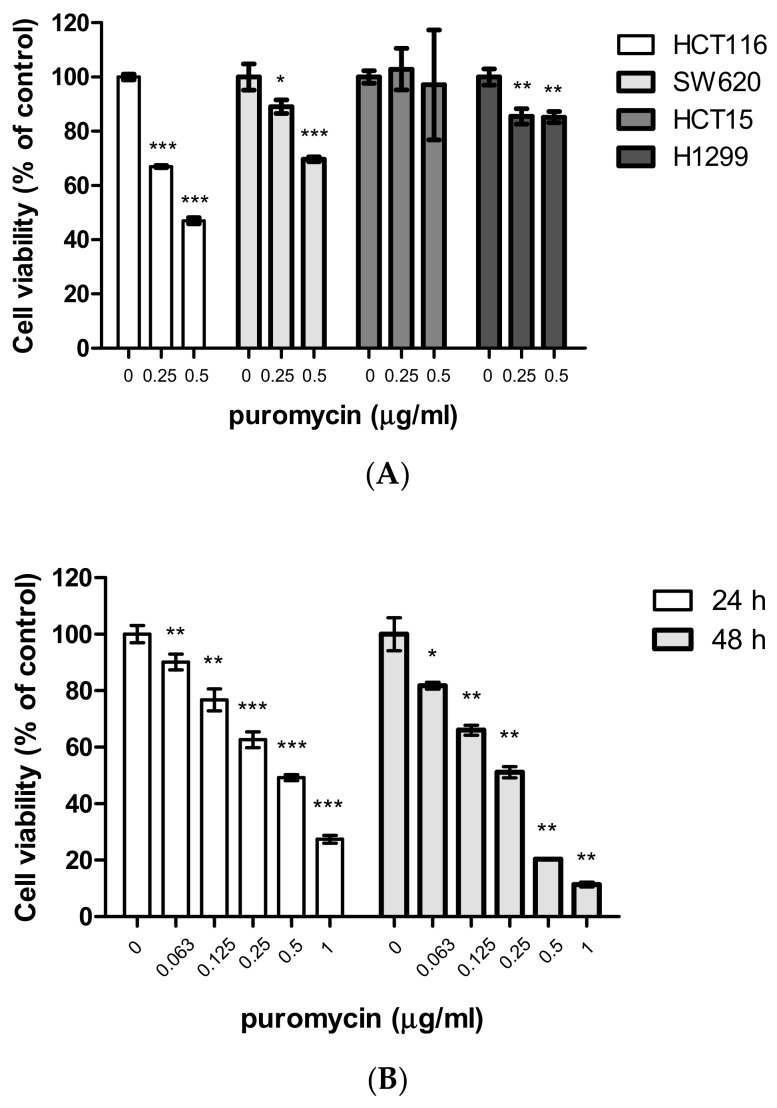
Puromycin exerts cytotoxic and antiproliferative effects in cancer cells. (**A**) Cytotoxicity of puromycin in HCT116, SW620, HCT15, and H1299 cells in a concentration-dependent manner by 3-[4,5-dimethylthiazole-2-yl]-2,5-diphenyltetrazolium bromide (MTT) assay. Data represent means ± SD. *, *p* < 0.05, **, *p* < 0.01, ***, *p* < 0.001 vs. untreated control. (**B**) Cytotoxicity of puromycin in HCT116 cells in a time-dependent manner by MTT assay. Data represent means ± SD. *, *p* < 0.05, **, *p* < 0.01, ***, *p* < 0.001 vs. untreated control. (**C**) Photos for colony formation (left) and bar graph (right) for colony formation in puromycin (0.5 μg/mL)-treated HCT116, H1299, SW620, and HCT15 cells. The colonies were visualized by staining with crystal violet and counted at OD590 nm. Data represent means ± SD. **, *p* < 0.01, ***, *p* < 0.001 vs. untreated control.

**Figure 2 cancers-11-00582-f002:**
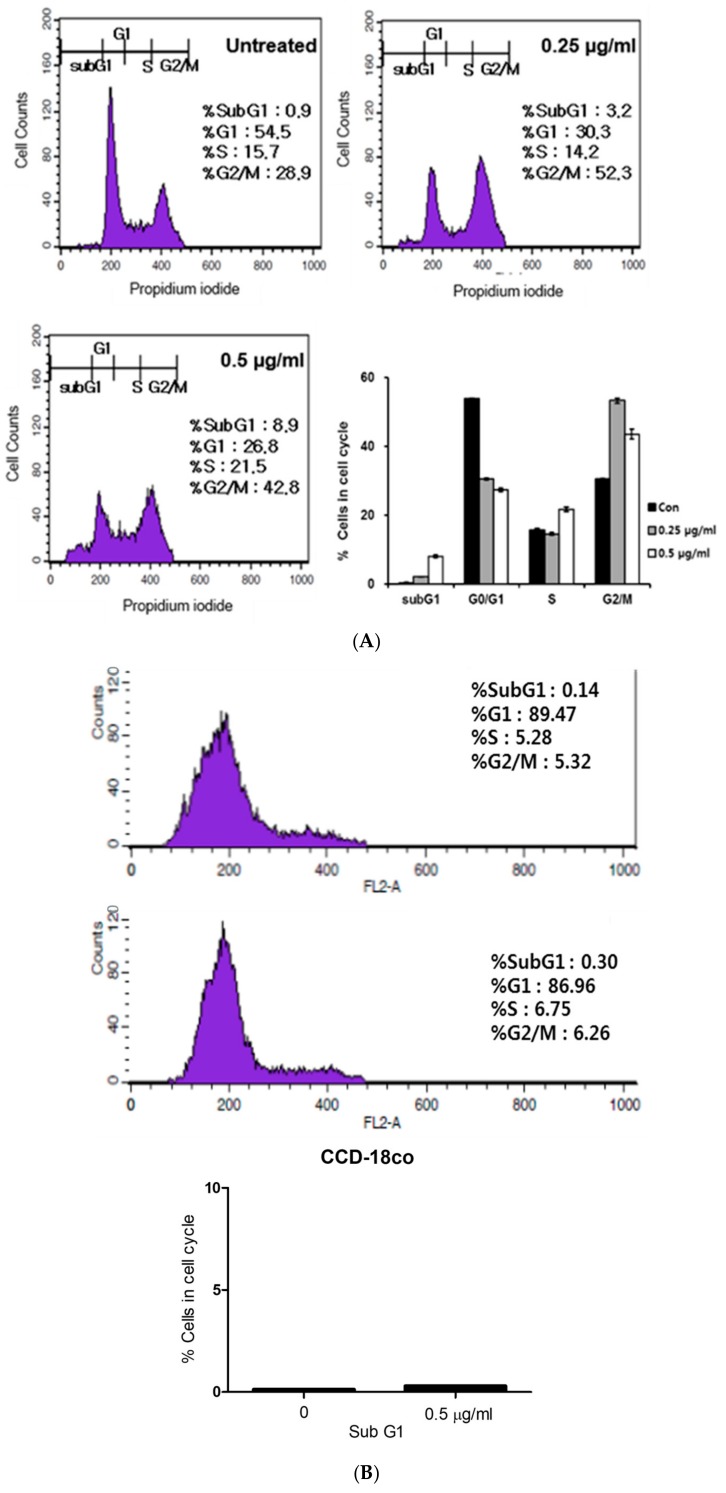
Puromycin increases the sub-G1 population and regulates apoptosis-related proteins in cancer cells. (**A**) Effect of puromycin (0.25 and 0.5 μg/mL) on cell cycle distribution in HCT116 cells by Fluorescence-activated cell sorting (FACS). (**B**) Effect of puromycin (0.5 μg/mL) on cell cycle distribution in CCD-18co cells by FACS. (**C**) Effect of puromycin on cyclin D1 and CDK4 in a concentration-dependent manner in HCT116 and H1299 cells. Cells were treated with puromycin (0.25 or 0.5 μg/mL) for 24 h and subjected to Western blotting with antibodies of cyclin D1, CDK4 and β actin. (**D**) Effect of puromycin on cyclin D1 and CDK4 in a time-dependent manner in HCT116 and H1299 cells. Cells were treated with 0.5 μg/mL puromycin for 3, 6, 12 or 24 h and subjected to Western blotting with antibodies of cyclin D1, CDK4, and β-actin. (**E**) Effect of puromycin on Poly (ADP-ribose) polymerase (PARP), Bcl-x_L_, Bcl-2, AKT, and p-AKT (v-akt murine thymoma viral oncogene homolog) in a concentration-dependent manner in HCT116 and H1299 cells. Cells were treated with puromycin (0.25 or 0.5 μg/mL) for 24 h and subjected to Western blotting with antibodies of PARP, Bcl-x_L_, Bcl-2, p-AKT, AKT, and β-actin. (**F**) Effect of puromycin on PARP, Bcl-xL, Bcl-2, AKT, and p-AKT in a time-dependent manner in HCT116 and H1299 cells. Cells were treated with 0.5 μg/mL of puromycin for 3, 6, 12 or 24 h and subjected to Western blotting with antibodies of PARP, Bcl-xL, Bcl-2, p-AKT, AKT, and β-actin.

**Figure 3 cancers-11-00582-f003:**
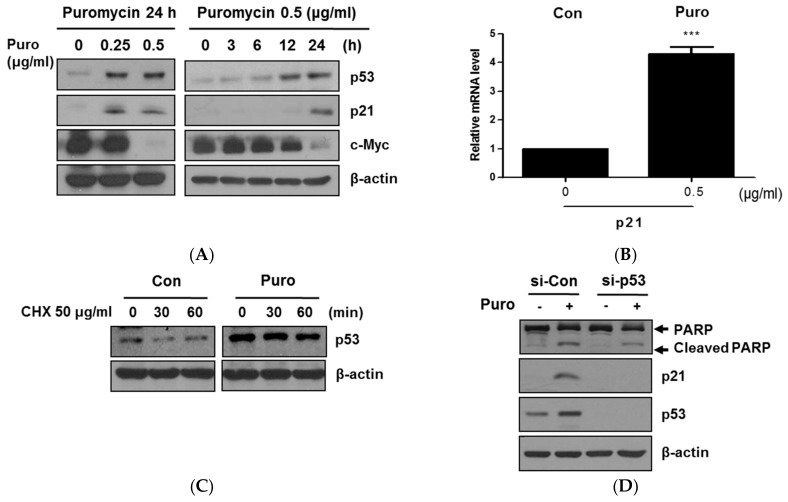
Puromycin induced p53-dependent apoptosis in HCT116 cells. (**A**) Effect of puromycin on p53, p21, and c-Myc in a concentration and time-dependent manner in HCT116 cells. HCT116 cells were exposed to various concentrations of puromycin (0.25 or 0.5 μg/mL) for 24 h and then subjected to Western blotting with antibodies of p53, p21, and c-Myc. Also, HCT116 cells were treated with 0.5 μg/mL puromycin for 6, 12, or 24 h and subjected to Western blotting. (**B**) Effect of puromycin on p21 in HCT116 cells by Reverse transcription polymerase chain reaction (RT-PCR). Total RNA from 0.5 μg/mL puromycin-treated HCT116 cells was isolated and subjected to RT-qPCR to analyze mRNA level of p21. ***, *p* < 0.001 vs. untreated control. (**C**) Effect of puromycin on p53 stability in HCT116 cells. HCT116 cells were exposed to 0.5 μg/mL puromycin for 24 h, with or without 50 μg/mL of protein synthesis inhibitor cycloheximide (CHX) for 30 and 60 min, and then subjected to Western blotting. (**D**) Effect of p53 depletion on PARP, p53, and p21 in puromycin-treated HCT116 cells. HCT116 cells were transfected with p53 small interfering RNA (siRNA) plasmid for 48 h and treated with 0.5 μg/mL of puromycin for 24 h. Western blotting was performed in puromycin-treated HCT116 cells with antibodies of PARP, p21, and p53. β-actin was used as an internal standard.

**Figure 4 cancers-11-00582-f004:**
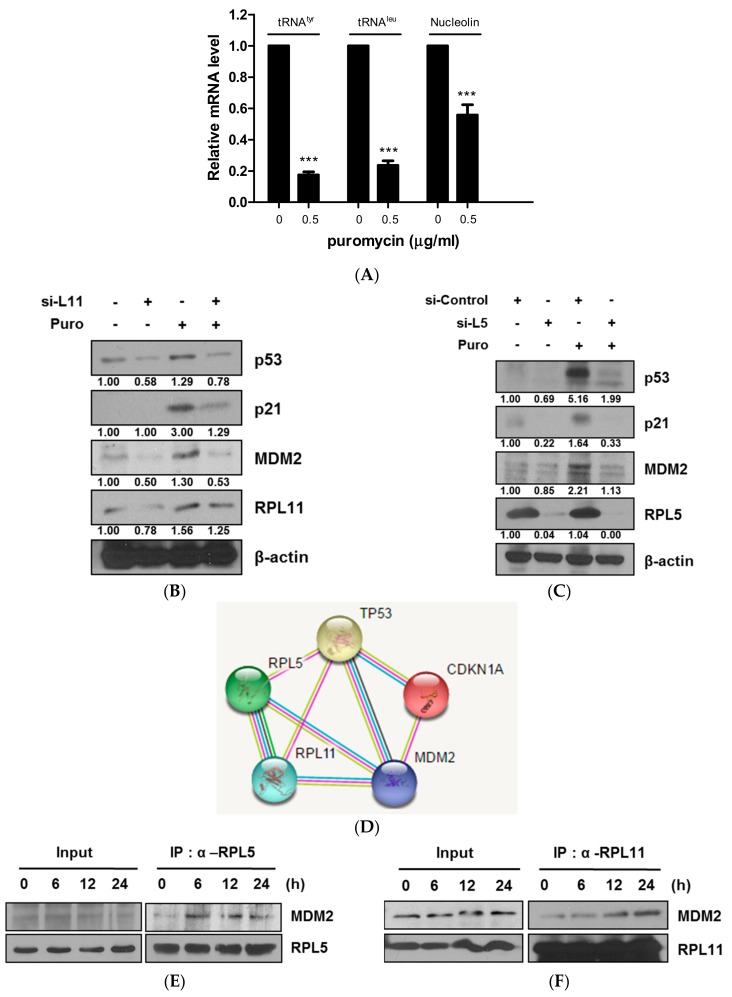
RPL5 or RPL11 mediates p53-dependent apoptosis and also directly binds to Mouse double minute 2 homolog (MDM2) in puromycin-treated HCT116 cells. (**A**) Effect of puromycin on p21 in HCT116 cells by RT-PCR. HCT116 cells were treated with puromycin (0 or 0.5 μg/mL) for 24 h. Total RNA was isolated and subjected to RT-qPCR to analyze mRNA levels of tRNATyr, tRNALeu, and nucleolin. (**B**) Effect of RPL11 depletion on p53, p2, RPL111, and MDM2 in HCT116 cells. HCT116 cells were transfected with control or L11 siRNA plasmid for 48 h, exposed to puromycin (0.5 μg/mL) for 24 h, and subjected to Western blotting with antibodies of p53, p21, MDM2, RPL11, and β-actin. (**C**) Effect of RPL5 depletion on p53, p21, RPL5, and MDM2 in HCT116 cells. HCT116 cells were transfected with control or L5 siRNA plasmid for 48 h, exposed to puromycin (0.5 μg/mL) for 24 h, and subjected to Western blotting with antibodies of p53, p21, MDM2, RPL5, and β-actin. (**D**) Binding scores for protein–protein interactions between TP53 (p53), CDKN1A (p21), MDM2, RPL5, and RPL11 by STRING v10.5. (**E**,**F**) Direct binding of RPL5 or RPL11 to MDM2 in puromycin-treated HCT116 cells by immunoprecipitation. HCT116 cells were treated with 0.5 μg/mL puromycin for different amounts of time. The whole cell lysates were immunoprecipitated using an anti-RPL5 or RPL11 with MDM2 by using immunoprecipitation.

**Figure 5 cancers-11-00582-f005:**
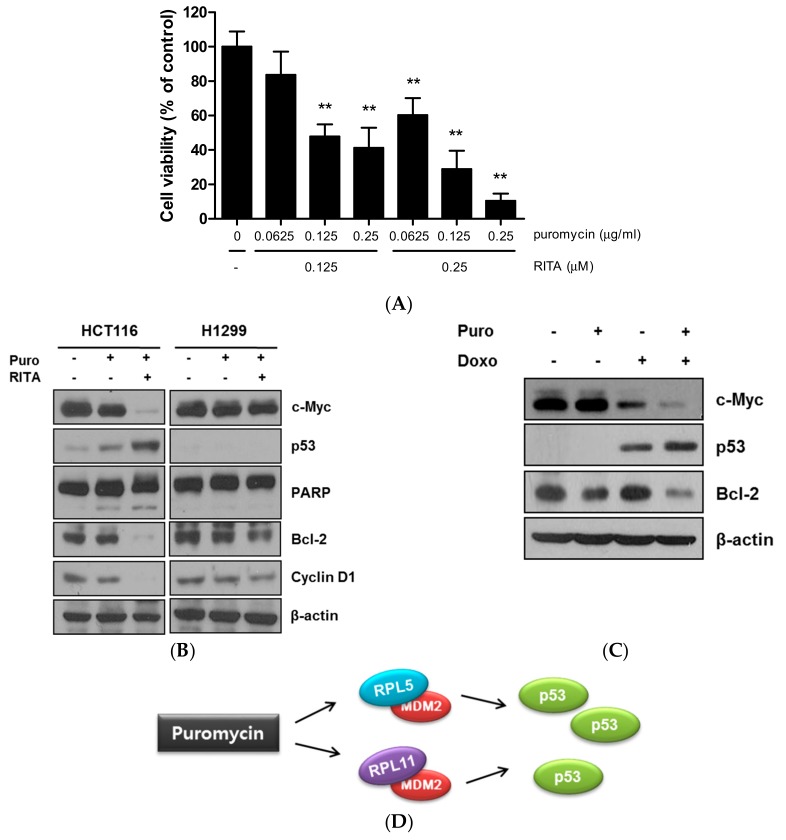
Combination effect of puromycin and reactivating p53 and inducing tumor apoptosis (RITA) or doxorubicin in cancer cells. (**A**) Synergistic effect of puromycin and RITA on cytotoxicity in HCT116 cells. HCT116 cells were treated with various concentrations of puromycin and RITA for 24 h, and cytotoxicity was evaluated by using MTT assay. **, *p* < 0.001 vs. untreated control. (**B**) Combination effect of puromycin and RITA on c-Myc, p53, PARP, Bcl-2, and cyclin D1 in HCT116 and H1299 cells. HCT116 and H1299 cells were treated with or without puromycin or/and RITA for 24 h. Western blotting was performed with antibodies of c-Myc, p53, PARP, Bcl-2, cyclin D1, and β-actin. (**C**) Combination effect of puromycin and doxorubicin on c-Myc, p53 and Bcl-2 in HCT116 cells. HCT116 cells were treated with or without puromycin (0.25 μg/mL) and doxorubicin (1 μM) for 24 h and subjected to Western blotting for c-Myc, p53, Bcl-2, and β-actin. (**D**) A mechanistic scheme for puromycin-induced p53 dependent apoptosis via interaction between MDM2 and RPL5/RPL11.

**Table 1 cancers-11-00582-t001:** Oligonucleotide sequences.

Primer Name	Sequences 5′→3′
p21-Forward	TCCAGGTTCAACCCACAGCTACTT
p21-Reverse	TCAGATGACTCTGGGAAACGCCAA
tRNATyr-Forward	CCTTCGATAGCTCAGCTGGTAGAGCGGAGG
tRNATyr-Reverse	CGGAATCGGAACCAGCGACCTAAGGATGTCC
tRNALeu-Forward	GTCAGGATGGCCGAGTGGTCTAAG
tRNALeu-Reverse	CCACGCCTCCATACGGAGAACCAGAAGACCC
Nucleolin-Forward	AAGCAGCACCTGGAAAACG
Nucleolin-reverse	TCTGAGCCTTCTACTTTCTGTTTCTTG
GAPDH-Forward	CACAATGCCGAAGTGGTCGT
GAPDH-Reverse	CACAATGCCGAAGTGGTCGT

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
