# Peer review of "p53-Dependent Apoptotic Effect of Puromycin via Binding of Ribosomal Protein L5 and L11 to MDM2 and Its Combination Effect with RITA or Doxorubicin"

_cancers, 2019, doi:10.3390/cancers11040582_

Round 1

Reviewer 1 Report

Thank you for addressing the raised concerns.

Reviewer 2 Report

I think that this revised manuscript is now suitable for publication in the Cancers.

This manuscript is a resubmission of an earlier submission. The following is a list of the peer review reports and author responses from that submission.

Round 1

Reviewer 1 Report

The manuscript of the article “p53 Dependent Apoptotic Effect of Puromycin via Binding of Ribosomal Protein L5 and L11 to MDM2 and its Synergistic Effect with RITA or Doxorubicin” by Jung et al. investigated the cytotoxic and anti-proliferative effects of puromycin treatment, mostly on HCT116 cells and p53 null H1299  cancer cell lines. The findings suggest that single treatment of  puromycin leading to stabilization of p53 can be used effectively in p53-proficient cancers, or more potent in combination with RITA or doxorubicin, to induce apoptosis via a mechanism involving MDM2 as well as RPL5 and/or RPL11. This is certainly a less investigated anti-cancer therapeutic approach which will be of interest to the readership of Cancers, however several concerns have to be addressed before the manuscript can be published.

Major concerns:

It should be explained of which types of cancer the individually used cell lines were derived from and why these cells were chosen for this study.

The abbreviation RITA needs to be defined and/ or explained in abstract, manuscript and legends for the unexperienced reader.

Fig.1a: It was stated that the viability of the HCT15 cancer cell line is significantly downregulated whereas HT1299 was not suppressed by puromycin treatment using MTT analysis which is inconsistent with the depicted graph. SW620 cells and HCT15 cells should also be integrated into the colony formation assay in Fig.1c. No incubation time and seeded cell number/well was indicated for the MTT assay in Fig.1a (should be consistent with the colony formation time of 24h), and used controls were not specified in Material and Methods.

Fig 2d: The amount of cleaved PARP in Fig. 2d does not correlate to the amount of cleaved PARP in Fig. 2e for the cell line H1299 after 24 h. Furthermore, compared to the untreated control the bands for Bcl-XL and Bcl-2 seem to be downregulated in Fig. 2d but not for Fig. 2e in the H1299 cell line after 24 h. Therefore this experiment needs to be repeated.

For the quantification of Western blots, it is unclear why in Fig. 4b the si-control/puromycin double-treated was chosen for standardization (i.e. 1.0) in contrast to the single si-control treated sample in Fig.4c.

Synergistic effects of combinations of drugs used in this study (i.e. puromycin and RITA or puromycin and doxorubicin) need to be assessed by using an additive model for example (ref.: 40. Valeriote F, Lin H (1975) Synergistic interaction of anticancer agents: a cellular perspective. Cancer Chemother Rep 59: 895–900.), or by other valid methods before the mode of treatment can be called “synergistic”.

Fig.5b and Fig.5c: The statement that synergistic treatment of HCT116 cells with puromycin and RITA effected these cells “more” than p53 null H1299 cells is not valid as  no deviation of H1299 Western blot bands  from the control can be identified. The same is true for Fig.5c where synergistic treatment of puromycin and doxorubicin was performed as no Western blots are shown for H1299 cells. It is unclear why synergistic effects were not shown for combined puromycin and doxorubicin treatment in H1299 cells

Materials and Methods are very condensed and several details are missing (e.g used controls, used dilutions of antibodies, percentages of SDS gels, RT-PCR amplification, evaluation method of primers, calculation method for RT-PCR, etc.)

Minor concerns:

The caption of Fig 1c is missing

Results, p.2 Fig.1: there is a typo in HCT15 cells (“HT15”)

For cell culture no incubation conditions were indicated in Material and Methods

Results 2.2: There is a typo in G1 population (“populstion”)

The x-axis of Fig.3b was labeled with “C” instead of “0” µg/ml puromycin

CHX needs to be explained in the legend of Fig. 3

The last paragraph (last two sentences) of the Discussion and the last sentence of Conclusions are remnants from the MDPI template and need to be deleted.

Author Response

Comments and Suggestions for Authors

The manuscript of the article “p53 Dependent Apoptotic Effect of Puromycin via Binding of Ribosomal Protein L5 and L11 to MDM2 and its Synergistic Effect with RITA or Doxorubicin” by Jung et al. investigated the cytotoxic and anti-proliferative effects of puromycin treatment, mostly on HCT116 cells and p53 null H1299 cancer cell lines. The findings suggest that single treatment of puromycin leading to stabilization of p53 can be used effectively in p53-proficient cancers, or more potent in combination with RITA or doxorubicin, to induce apoptosis via a mechanism involving MDM2 as well as RPL5 and/or RPL11. This is certainly a less investigated anti-cancer therapeutic approach which will be of interest to the readership of Cancers, however several concerns have to be addressed before the manuscript can be published.

Major concerns:

It should be explained of which types of cancer the individually used cell lines were derived from and why these cells were chosen for this study.

Response: Thanks. Added in Cell Culture section and Introduction.

The abbreviation RITA needs to be defined and/ or explained in abstract, manuscript and legends for the unexperienced reader.

Response: Defined.

Fig.1a: It was stated that the viability of the HCT15 cancer cell line is significantly downregulated whereas HT1299 was not suppressed by puromycin treatment using MTT analysis which is inconsistent with the depicted graph. SW620 cells and HCT15 cells should also be integrated into the colony formation assay in Fig.1c. No incubation time and seeded cell number/well was indicated for the MTT assay in Fig.1a (should be consistent with the colony formation time of 24h), and used controls were not specified in Material and Methods.

Response: Thanks. Colony formation data in SW620 cells and HCT15 cells were added in Fig.1c

Fig 2d: The amount of cleaved PARP in Fig. 2d does not correlate to the amount of cleaved PARP in Fig. 2e for the cell line H1299 after 24 h. Furthermore, compared to the untreated control the bands for Bcl-XL and Bcl-2 seem to be downregulated in Fig. 2d but not for Fig. 2e in the H1299 cell line after 24 h. Therefore this experiment needs to be repeated.

Response: Thanks. New blots were added in Fig. 2e.

For the quantification of Western blots, it is unclear why in Fig. 4b the si-control/puromycin double-treated was chosen for standardization (i.e. 1.0) in contrast to the single si-control treated sample in Fig.4c.

Response: Thanks. Corrected as in Fig4c.

Synergistic effects of combinations of drugs used in this study (i.e. puromycin and RITA or puromycin and doxorubicin) need to be assessed by using an additive model for example (ref.: 40. Valeriote F, Lin H (1975) Synergistic interaction of anticancer agents: a cellular perspective. Cancer Chemother Rep 59: 895–900.), or by other valid methods before the mode of treatment can be called “synergistic”.

Response: Thanks. We removed “synergistic” effect by combination of puromycin and doxorubiin and used “enhanced antitumor effect of RIRA or doxorubicin”. Also, Title was changed as “p53 Dependent Apoptotic Effect of Puromycin via Binding of Ribosomal Protein L5 and L11 to MDM2 and its Combination Effect with RITA or Doxorubicin”

Fig.5b and Fig.5c: The statement that synergistic treatment of HCT116 cells with puromycin and RITA effected these cells “more” than p53 null H1299 cells is not valid as no deviation of H1299 Western blot bands from the control can be identified. The same is true for Fig.5c where synergistic treatment of puromycin and doxorubicin was performed as no Western blots are shown for H1299 cells. It is unclear why synergistic effects were not shown for combined puromycin and doxorubicin treatment in H1299 cells

Response: Thanks. Synergistic was removed in Results.

Materials and Methods are very condensed and several details are missing (e.g used controls, used dilutions of antibodies, percentages of SDS gels, RT-PCR amplification, evaluation method of primers, calculation method for RT-PCR, etc.)

Minor concerns:

The caption of Fig 1c is missing

Response: Added.

Results, p.2 Fig.1: there is a typo in HCT15 cells (“HT15”)

Response: Corrected.

For cell culture no incubation conditions were indicated in Material and Methods

Response: Added.

Results 2.2: There is a typo in G1 population (“populstion”)

Response: Corrected.

The x-axis of Fig.3b was labeled with “C” instead of “0” µg/ml puromycin

Response: Corrected.

CHX needs to be explained in the legend of Fig. 3

Response: Added as protein synthesis inhibitor

The last paragraph (last two sentences) of the Discussion and the last sentence of Conclusions are remnants from the MDPI template and need to be deleted.

Response: Removed.

Reviewer 2 Report

Review Report

Cancers 446572 article

Title:  p53 Dependent Apoptotic Effect of Puromycin via Binding of Ribosomal Protein L5 and L11 to MDM2 3 and its Synergistic Effect with RITA or Doxorubicin

Authors: Ji Hoon Jung, Hyemin Lee and Sung-Hoon Kim

Jung et al provide evidence for p53 reactivation and induction of apoptosis by puromycin via binding of RPL5 and L11 to the p53 inhibitor MDM2, and evidence for synergistic effects of combining puromycin with the p53 activating compounds RITA or doxorubicin to induce cell cycle arrest and apoptosis in p53wt solid cancer cell lines. 

The content of the article appears to be adequate for the description of the stated phenomenon in both the quality and amount of data. The discussion section is rather poor, in comparison, as if written by a different author with a lack of eloquence and articulation.

Specific comments

The abstract, introduction, data and results section are of good quality, the discussion section is rather poor.

1)      Line 28: Most scientist use puromycin for selection of cell lines only, not for clinical therapies.

It would be helpful to provide more basic information on puromycin, and on the development of puromycin derivatives for clinical applications.

Puromycin is a structural analog of tyrosyl tRNA (Reference:  tRNA mimics by Giegé, Frugier and Rudinger - Current opinion in structural biology, 1998). The structural similarities between puromycin and the aminoacyl adenyl terminal of aminoacyl-tRNA allow it to terminate the protein synthesis process by causing premature release of a polypeptide from the ribosome in normal and diseased cells. Interestingly, compared to normal cells, malignant cells have higher sensitivity to the absence of specific amino acids, leaving them vulnerable to apoptosis and reduced proliferation (Scott et al., Br. J. Cancer 83 (2000) 800e810).

2)      Figure 1. The authors show that puromycin treatment is effective in malignant p53wt cells, but not in malignant p53mutant cells. There is, however, a significant effect in malignant p53null cells, H1299 (Fig. 1a and 1d). This has not been discussed by the authors.

Moreover, there is a high risk that puromycin would have the same effect on healthy cells. In an in vitro study undertaken in a variety of malignant and non-malignant human and murine cell types low doses of puromycin (0.1–0.5 μg/ml) disrupted significant phase-to-phase cell cycle transitions, causing a G2-arrest, a metaphase-mitotic-arrest, and apoptosis (Davidoff and Mendelow, Leuk Res 16, 1992, 1077-1085).

Therefore, the authors have to include data on puromycin treatment in non-malignant human cells, e.g. normal colon CCD841-CoN cells.

3)      Line 230-233: Remove the instructions. The discussion section is rather poor.

Puromycin is not used in cancer therapy. Puromycin derivatives have been synthesized and studied for cancer therapy (Singh et al., 2017; Ueki et al., 2016).  This should be discussed.

Relevant refences:  Singh, Williams, Vince. Eur J Med Chem. 2017 Oct 20;139:325-336. doi: 10.1016/ j.ejmech. 2017.07.048. Puromycin based inhibitors of aminopeptidases for the potential treatment of hematologic malignancies.

Ueki et al., Theranostics. 2016 Mar 28;6(6):808-16. doi: 10.7150/thno.13826. eCollection 2016. Synthesis and Preclinical Evaluation of a Highly Improved Anticancer Prodrug Activated by Histone Deacetylases and Cathepsin L.

4)      Line 215: Also, it was well known that ribosomal protein L5 [30], 215 L11 [31], and L23[32] target MDM2 and p53 in response to ribosomal stress [9, 21], since these proteins directly interact to MDM2 to block p53 ubiquitination mediated by MDM2 [27, 33, 34].

The relevant references for these statements should be cited:

Zhang, Y., Wolf, G. W., Bhat, K., Jin, A., Allio, T., Burkhart, W. A., and Xiong, Y. (2003) Mol. Cell. Biol. 23, 8902–8912

Lohrum, M. A., Ludwig, R. L., Kubbutat, M. H., Hanlon, M., and Vousden, K. H. (2003) Cancer Cell 3, 577–587

Jin, A., Itahana, K., O'Keefe, K., and Zhang, Y. (2004) Mol. Cell. Biol. 24, 7669–7680

Dai, M. S., and Lu, H. (2004) J. Biol. Chem. 279, 44475–44482

Dai, M. S., Zeng, S. X., Jin, Y., Sun, X. X., David, L., and Lu, H. (2004) Mol. Cell. Biol. 24, 7654–7668

Bhat, K. P., Itahana, K., Jin, A., and Zhang, Y. (2004) EMBO J. 23, 2402–2412

Line, grammar corrections, minor issues:

1)      Line 190:    Herein puromycin significantly showed significant cytotoxicity in .… word redundancy

My proposal: Puromycin treatment induced significant loss of cell viability…

2)      Line 195: References are not relevant. Information is not relevant. “Apoptotic portion” is not a correct english expression. Omit the sentence.

It is well documented that increase of sub G1 phase by perturbing cell cycle progression is recognized as an apoptotic portion [24, 25].

3)      Line 215: interact with MDM2 instead of interact to MDM2

4)      Line 245: It should be mentioned which puromycin was used and where it was purchased.

5)      Line 305.  p53/p21 in in p53 wild type…. word duplication

6)      Line 309: Remove the instructions.

Author Response

Reviewer 2’s Comments and Suggestions for Authors

Jung et al provide evidence for p53 reactivation and induction of apoptosis by puromycin via binding of RPL5 and L11 to the p53 inhibitor MDM2, and evidence for synergistic effects of combining puromycin with the p53 activating compounds RITA or doxorubicin to induce cell cycle arrest and apoptosis in p53wt solid cancer cell lines.

The content of the article appears to be adequate for the description of the stated phenomenon in both the quality and amount of data. The discussion section is rather poor, in comparison, as if written by a different author with a lack of eloquence and articulation.

Specific comments

The abstract, introduction, data and results section are of good quality, the discussion section is rather poor.

Response: Thanks. Improved

1). Line 28: Most scientist use puromycin for selection of cell lines only, not for clinical therapies.

It would be helpful to provide more basic information on puromycin, and on the development of puromycin derivatives for clinical applications.

Puromycin is a structural analog of tyrosyl tRNA (Reference: tRNA mimics by Giegé, Frugier and Rudinger - Current opinion in structural biology, 1998). The structural similarities between puromycin and the aminoacyl adenyl terminal of aminoacyl-tRNA allow it to terminate the protein synthesis process by causing premature release of a polypeptide from the ribosome in normal and diseased cells. Interestingly, compared to normal cells, malignant cells have higher sensitivity to the absence of specific amino acids, leaving them vulnerable to apoptosis and reduced proliferation (Scott et al., Br. J. Cancer 83 (2000) 800e810).

Response: Thanks. Added in Introduction with references based on your comments.

2). Figure 1. The authors show that puromycin treatment is effective in malignant p53wt cells, but not in malignant p53mutant cells. There is, however, a significant effect in malignant p53null cells, H1299 (Fig. 1a and 1d). This has not been discussed by the authors.

Response: Added in Discussion

Moreover, there is a high risk that puromycin would have the same effect on healthy cells. In an in vitro study undertaken in a variety of malignant and non-malignant human and murine cell types low doses of puromycin (0.1–0.5 μg/ml) disrupted significant phase-to-phase cell cycle transitions, causing a G2-arrest, a metaphase-mitotic-arrest, and apoptosis (Davidoff and Mendelow, Leuk Res 16, 1992, 1077-1085). Therefore, the authors have to include data on puromycin treatment in non-malignant human cells, e.g. normal colon CCD841-CoN cells.

Response: Cell cycle analysis revealed no toxicity of puromycin(0.5 μg/ml) in normal colon CCD841-CoN cells (Figure 2b).

3). Line 230-233: Remove the instructions. The discussion section is rather poor.

Puromycin is not used in cancer therapy. Puromycin derivatives have been synthesized and studied for cancer therapy (Singh et al., 2017; Ueki et al., 2016). This should be discussed.

Relevant refences: Singh, Williams, Vince. Eur J Med Chem. 2017 Oct 20;139:325-336. doi: 10.1016/ j.ejmech. 2017.07.048. Puromycin based inhibitors of aminopeptidases for the potential treatment of hematologic malignancies.

Ueki et al., Theranostics. 2016 Mar 28;6(6):808-16. doi: 10.7150/thno.13826. eCollection 2016. Synthesis and Preclinical Evaluation of a Highly Improved Anticancer Prodrug Activated by Histone Deacetylases and Cathepsin L.

Response: The instructions were removed and above references were cited.

4). Line 215: Also, it was well known that ribosomal protein L5 [30], 215 L11 [31], and L23[32] target MDM2 and p53 in response to ribosomal stress [9, 21], since these proteins directly interact to MDM2 to block p53 ubiquitination mediated by MDM2 [27, 33, 34].

The relevant references for these statements should be cited:

Zhang, Y., Wolf, G. W., Bhat, K., Jin, A., Allio, T., Burkhart, W. A., and Xiong, Y. (2003) Mol. Cell. Biol. 23, 8902–8912

Lohrum, M. A., Ludwig, R. L., Kubbutat, M. H., Hanlon, M., and Vousden, K. H. (2003) Cancer Cell 3, 577–587

Jin, A., Itahana, K., O'Keefe, K., and Zhang, Y. (2004) Mol. Cell. Biol. 24, 7669–7680

Dai, M. S., and Lu, H. (2004) J. Biol. Chem. 279, 44475–44482

Dai, M. S., Zeng, S. X., Jin, Y., Sun, X. X., David, L., and Lu, H. (2004) Mol. Cell. Biol. 24, 7654–7668

Bhat, K. P., Itahana, K., Jin, A., and Zhang, Y. (2004) EMBO J. 23, 2402–2412

Response: Thanks. Cited.

Line, grammar corrections, minor issues:

1). Line 190: Herein puromycin significantly showed significant cytotoxicity in .… word redundancy

My proposal: Puromycin treatment induced significant loss of cell viability…

Response: Corrected

2). Line 195: References are not relevant. Information is not relevant. “Apoptotic portion” is not a correct english expression. Omit the sentence. It is well documented that increase of sub G1 phase by perturbing cell cycle progression is recognized as an apoptotic portion [24, 25].

Response: Corrected

3). Line 215: interact with MDM2 instead of interact to MDM2

Response: Corrected

4). Line 245: It should be mentioned which puromycin was used and where it was purchased.

Response: Added

5). Line 305. p53/p21 in in p53 wild type…. word duplication

Response: Corrected

6). Line 309: Remove the instructions.

Response: Removed